# Localization of Plastic Deformation in Ti-6Al-4V Alloy

**Vladimir V. Skripnyak * and Vladimir A. Skripnyak**

Department of Mechanics of Deformable Solids, National Research Tomsk State University, 36 Lenin Av.,
634050 Tomsk, Russia; skrp2006@yandex.ru
* Correspondence: skrp2012@yandex.ru; Tel.: +7-923-402-77-44

**Abstract:** This article investigated the mechanical behavior of Ti-6Al-4V alloy (VT6, an analog to Ti Grade 5) in the range of strain rates from 0.1 to $10^3$ s$^{-1}$. Tensile tests with various notch geometries were performed using the Instron VHS 40/50-20 servo hydraulic testing machine. The Digital Image Correlation (DIC) analysis was employed to investigate the local strain fields in the gauge section of the specimen. The Keyence VHX-600D digital microscope was used to characterize full-scale fracture surfaces in terms of fractal dimension. At high strain rates, the analysis of the local strain fields revealed the presence of stationary localized shear bands at the initial stages of strain hardening. The magnitude of plastic strain within the localization bands was significantly higher than those averaged over the gauge section. It was found that the ultimate strain to fracture in the zone of strain localization tended to increase with the strain rate. At the same time, the Ti-6Al-4V alloy demonstrated a tendency to embrittlement at high stress triaxialities.

**Keywords:** localization of deformation; titanium alloys; stress triaxiality; high strain rate; fracture; topology of fracture surface



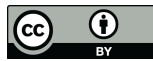

## 1. Introduction

The alloy Ti-6Al-4V (VT6) is widely used for the manufacture of light, reliable, and corrosion-resistant parts of mechanisms and machines, and structural elements of aerospace and transport systems [1]. Despite the large amount of research carried out, the design of 3D constructional elements with a complex geometric shape made of Ti-6Al-4V alloy is a serious scientific and technical problem, especially for the cases where the material suffers under extreme conditions [2]. It has been found that mechanical properties and failure mechanisms of Ti-6Al-4V depend on strain rate and stress triaxiality [3,4]. Due to its low heat conductivity, the Ti-6Al-4V undergoes localization of plastic deformation under high strain rates. The localization takes place by the formation of two-dimensional interfacial regions, which are commonly referred to as shear bands [5,6]. The formation and widening of shear bands is a complex phenomenon, which is influenced by mechanical properties of the material such as work hardening, strain rate sensitivity, and thermal softening [5,7]. These properties depend on the thermodynamic state of the material and its grain structure. Furthermore, it has been found that the internal structure of the material undergoes transformation within shear bands. In [8–10], results indicate the occurrence of localized melting within shear bands. The gradient of tensile deformation in these regions results in softening due to the temperature rise and/or nucleation of voids, leading to the further concentration of deformation [7–10]. Therefore, the similarity of regularities between the stress triaxiality and ultimate strain to fracture under quasi-static and dynamic loading conditions is questionable.

It should be noted, however, that the studies on the influence of stress triaxiality on shear instabilities taking place at high strain rates are not widely presented in the literature.

Tensile tests with notched geometries are utilized to investigate the influence of the constraint factor on the mechanical behavior of metals and alloys. The obtained data are used to calibrate constitutive models such as Johnson–Cook, Gurson–Tvergaard-Needleman,

GISSMO, and other pressure-dependent damage models [11–15]. As the assumption of uniform strain in the gauge section is not valid in these tests, often, model parameters cannot be determined in a straightforward manner. The correctness of determined parameters in this case is ensured by matching force–displacement curves from experiment and FEM simulation. Fracture characteristics are usually evaluated by examining post-tested specimens [14].

Therefore, obtaining more complete information on the laws of localization of plastic deformation remains an urgent problem. Its solution will help to improve the accuracy of reliability analysis for critical structures made of Ti-6Al-4V alloy and subjected to dynamic loads, as well as to obtain required mechanical properties of the alloy as a result of the purposeful change in its structure.

The aim of this work was to expand the understanding on the laws governing the development of plastic flow and deformation of the Ti-6Al-4V alloy under tension in the range of strain rates from 0.1 to $10^3$ s$^{-1}$.

It should also be noted that the range of strain rates under investigation is lower than those produced by the split Hopkinson bar technique ($10^4$ s$^{-1}$) and higher than those achieved on testing machines in quasi-static loading (up to $10^{-1}$ s$^{-1}$). At the same time, it is known that the critical strain rate to form adiabatic shear bands in Ti-6Al-4V is ~$10^3$ s$^{-1}$ [5,16].

From a practical point of view, the range of strain rates up to $10^3$ can be realized in a number of technological processes such as cutting, drilling, and in problems of automotive and aerospace crash-tests.

## 2. Materials and Methods

The regularities of the mechanical behavior of the Ti-6Al-4V alloy were investigated under tension in the range of strain rates from 0.1 to $10^3$ s$^{-1}$ and at room temperature. The microstructure survey and the determination of the chemical composition of the specimen were carried out using a Tescan Vega TS 5130 MM scanning electron microscope equipped with a LINK energy-dispersive spectrometer (Oxford Instruments). The alloy had a chemical composition in weight %: Ti~90.64; Al~5.95; V~3.31 and was in a polycrystalline state with an average grain size of ~25 μm. The structure of the Ti-6Al-4V alloy (see Figure 1a) consists of a combination of equiaxed grains of the alpha phase with a hexagonal close-packed (hcp) lattice and grains with a lamellar structure formed by plates of alpha and beta phases. The beta phase has a body-centered cubic (bcc) lattice.

The specimens were cut from thin-sheet rolled Ti-6Al-4V alloy by the electroerosion method. The sample thickness $d$ was $1.1 \pm 0.01$ mm, and the smallest width $w$ was 6 mm. The initial gauge length of the specimens was $20 \pm 0.1$ mm. Figure 1b shows the geometry of the titanium samples. The minimum cross-sectional area of flat specimens ($w \times d$) was $A_0 = 6.6 \pm 0.06$ mm$^2$.

Tests were carried out in accordance with ISO 26203-2: 2011 on the Instron VHS 40/50-20 servo-hydraulic test bench. The tensile force and displacements were recorded with high temporal resolution up to the fracture. For each type of specimen, three tests were carried out at each of the strain rates $10^3$, $10^2$, and 0.1 s$^{-1}$. In each series of tests, a high degree of reproducibility of strain rate, forces, and displacements was observed.

The local strain fields of the specimen were obtained by the Digital Image Correlation (DIC) method [17]. A Phantom V711 camera (Vision Research-AMETEK Co., Wayne, NJ, USA) with a speed of $10^5$ frames per second was used to record changes in the specimen geometry.

The video was recorded in several resolutions: $1280 \times 800$, $1024 \times 680$, and $512 \times 400$ pixels at strain rates of $10^3$, $10^2$, and 0.1 s$^{-1}$. The image size varied depending on the allowable resolution for high-speed shooting of deformable specimens. The dimensions of the image were used with 250 pixels along the minimum width of the gauge section of the specimen.

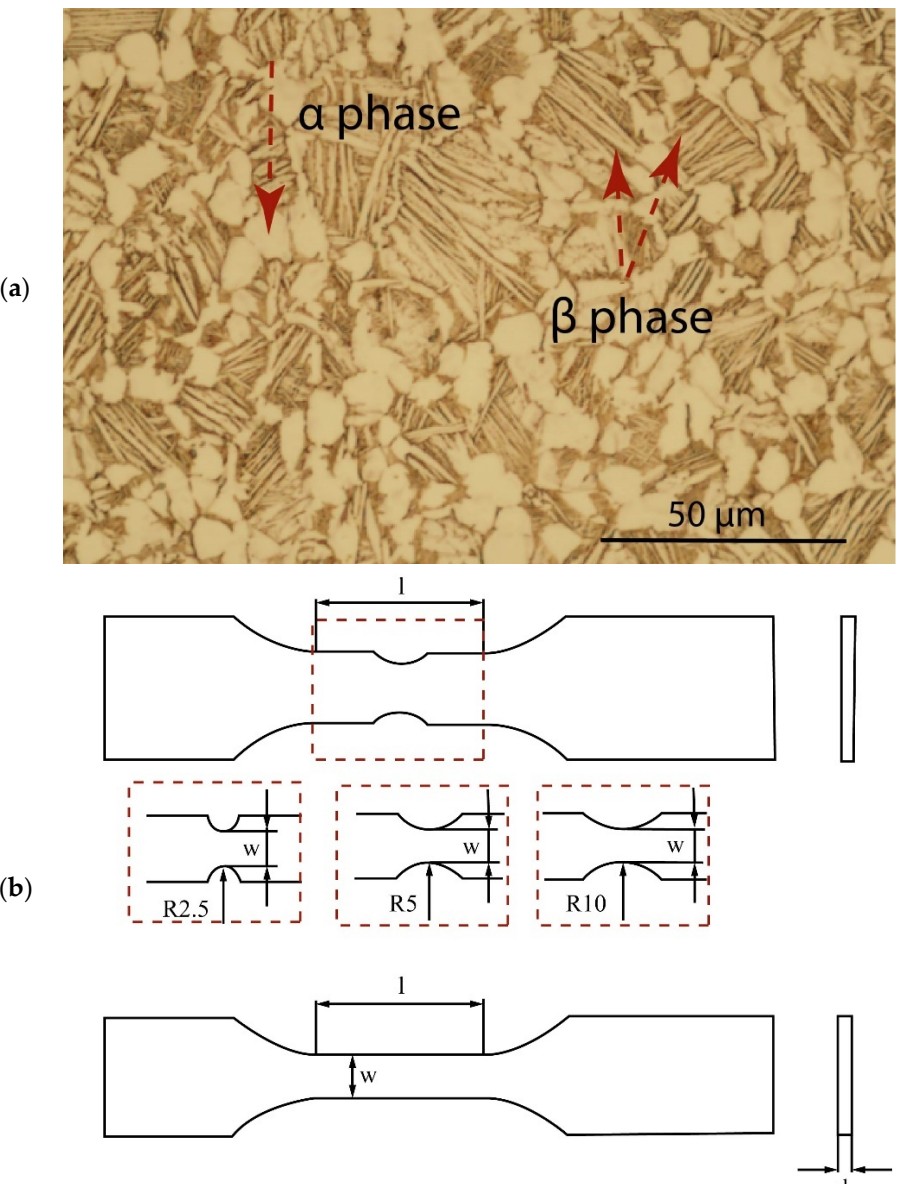

**Figure 1.** Microstructure of Ti-6Al-4V alloy (**a**); sample's geometry (**b**).

The analysis of strain fields by the DIC method enables one to select the size and position of the virtual extensometer on the gauge section of the specimen in the zone of localization of the plastic flow [18,19]. This makes it possible to increase the accuracy and adequacy of the obtained true stress–true strain diagrams on a stage of diffuse necking.

The size of the virtual extensometer was chosen so that the deformation was uniform along its length until the moment corresponding to the appearance of the final descending branch (pre-fracture) on the force–displacement diagram. Therefore, correct values of flow stress were obtained by taking into account the true-strain in the necking region of the specimen when flow localization was initiated.

True stress $\sigma_{1\,\text{true}}$ and true strain $\varepsilon_{1\,\text{true}}$ for unnotched specimens were determined by the formulas [20]:

$$\sigma_{1\,\text{true}} = (F/A_0)(1 - \Delta L/L_0), \tag{1}$$

$$\varepsilon_{1\,\text{true}} = \ln(1 - \Delta L/L_0), \tag{2}$$

where F is the tensile force, $A_0$ is initial minimal cross-sectional area, $\Delta L$ is the relative elongation of the virtual extensometer, and $L_0$ is the initial length of the virtual extensometer.

The stress triaxiality parameter η was determined by the formula [11]:

$$\eta = -\mathrm{p}/\sigma_{eq}, \tag{3}$$

where $\mathrm{p} = -(\sigma_{11} + \sigma_{22} + \sigma_{33})/3$ is the hydrostatic pressure, $\sigma_{eq} = \left[(3/2)\left(\sigma_{ij} - \mathrm{p}\delta_{ij}\right)\left(\sigma_{ij} - \mathrm{p}\delta_{ij}\right)\right]^{1/2}$ is the equivalent stress (von Mises), $\sigma_{ij}$ are components of the stress tensor, $\delta_{ij}$ is the Kronecker symbol, and i, j = 1, 2, 3.

The initial value of $\eta$ for notched flat specimens was calculated using the analytical formula for the plane stress state [21]:

$$\eta = (1 + 2\mathrm{A})/(3\sqrt{\mathrm{A}^2 + \mathrm{A} + 1}, \ \mathrm{A} = \ln[1 + w/(4R)], \tag{4}$$

where $w$ is minimal width of the specimen's gauge part, and $R$ is the notch radius.

For smooth specimen $\eta = 0.333$, for specimens with notch radii of 2.5, 5, and 10 mm, the initial values of $\eta$ were 0.4973, 0.4405, and 0.39612, respectively. The equivalent plastic strain $\varepsilon_{eq}^{p}$ in the case of the smooth specimen was determined by taking into account $\varepsilon_{2}^{p} = \varepsilon_{3}^{p} = -(1/2)\,\varepsilon_{1}^{p}$ by the formula:

$$\varepsilon_{1}^{p} = \varepsilon_{1}^{true} - \sigma_{1}^{true}/E, \tag{5}$$

$$\varepsilon_{eq}^{p} = (\sqrt{2}/3)[(\varepsilon_{1}^{p} - \varepsilon_{2}^{p})^2 + (\varepsilon_{2}^{p} - \varepsilon_{3}^{p})^2 + (\varepsilon_{3}^{p} - \varepsilon_{1}^{p})^2]^{1/2}, \tag{6}$$

where $E$ is the Young's modulus, and $\varepsilon_{i}^{p}$ are principal components of the plastic strain tensor.

The equivalent strain in the case of a uniaxial stress state was determined as:

$$\varepsilon_{eq} = (\sigma_{eq}/E) + \varepsilon_{eq}^{p}, \tag{7}$$

The average strain rate $\dot{\varepsilon}_1$ was determined by the formula [22]:

$$\dot{\varepsilon}_1 = v_1(t)/l, \tag{8}$$

where $v_1(t)$ is the loading velocity, $t$ is time, and $l$ is the length of the gauge section.

## 3. Results

### 3.1. Flow Stress and Ultimate Elongation before Fracture

The true stress–true strain diagrams of the Ti-6Al-4V alloy for strain rates of $10^3$, $10^2$, and 0.1 s$^{-1}$, obtained under uniaxial tension of smooth Ti-6Al-4V specimens ($\eta = 0.33$), are shown in Figure 2a.

The plastic flow stress and ultimate strain to fracture demonstrate the sensitivity to strain rate. The results showed that the degree of macroscopic homogeneous deformation in the specimens decreased with the strain rate (see Figure 2a).

The formation of a system of quasi-stationary localization bands was observed in the smooth specimens at strain rates above $10^2$ s$^{-1}$. Note that the elongations corresponding to the beginning of softening significantly exceeded those at which quasi-stationary localization bands were formed (see Figure 2a).

With the increase in the loading time, the magnitude of $\varepsilon_{1\mathrm{true}}$ increased within the localization bands, while the position of the bands on the gauge section remained constant. The localization led to local heating to a temperature significantly higher than the average temperature in the gauge section. Local heating in localized shear bands prevented the nucleation and growth of voids at the microlevel, and, as a result, the ultimate strain to fracture increased.

Figure 2b shows force–displacement diagrams for smooth and notched specimens. An increase in the tensile force at the effective strain rate of $10^3$ s$^{-1}$ was associated with the deceleration of the Luders fronts due to the appearance of an equivalent stress gradient in the region of the variable cross-section area of the specimen.

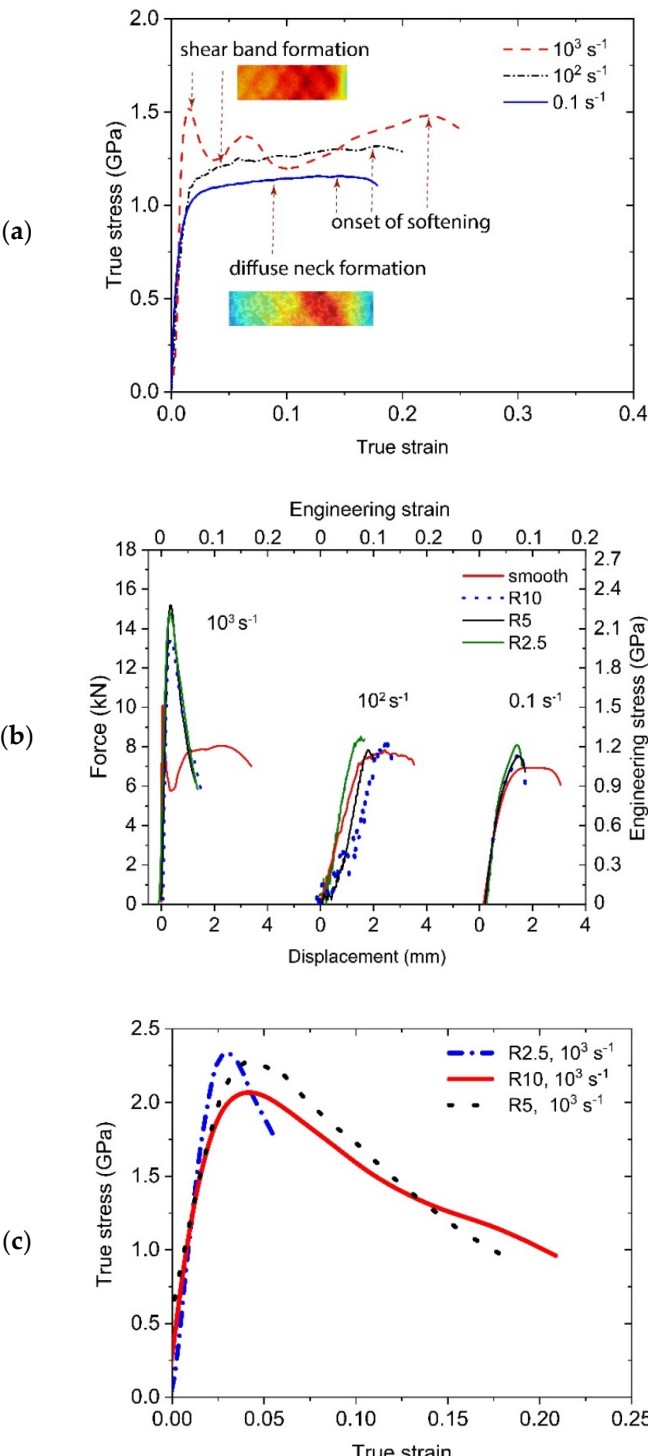

**Figure 2.** True stress versus true strain curves under tension of flat specimens (**a**) and force–displacement diagrams of Ti-6Al-4V at strain rates of $10^3$, $10^2$, and 0.1 s$^{-1}$ (**b**); true stress versus true strain curves under tension of notched specimens at $10^3$ s$^{-1}$ (**c**).

Figure 3 shows local fields of equivalent strain in the gauge section at the time moment preceding the onset of fracture. The results indicated that the ultimate strain to fracture depended on the type of stress state and strain rate.

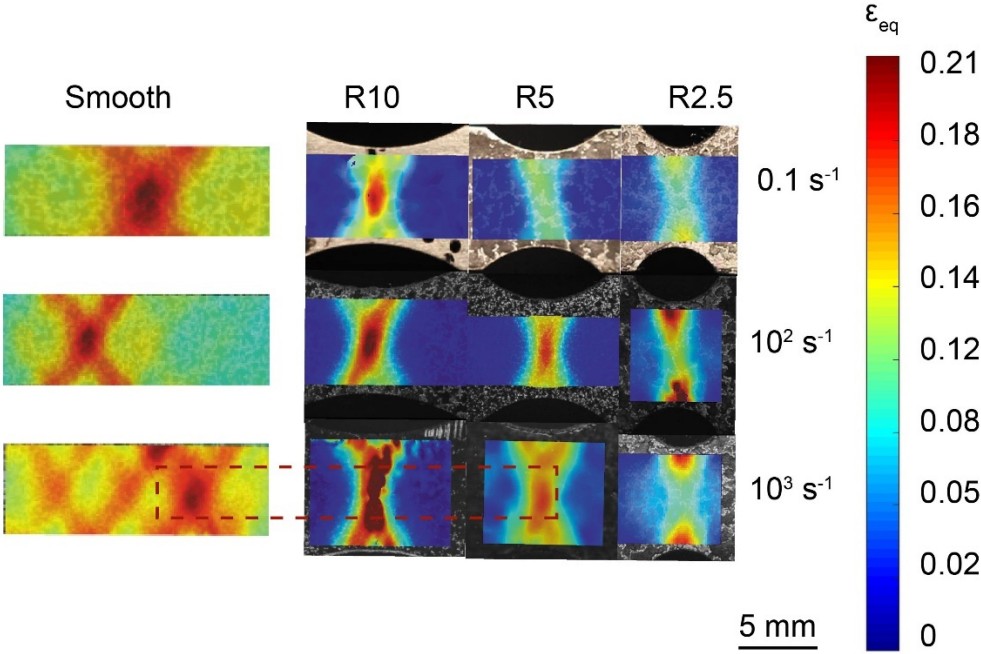

**Figure 3.** Equivalent strains in smooth specimens under tension with strain rates of $10^3$, $10^2$, and 0.1 s$^{-1}$ (top insert), and equivalent strains in specimens with notch radii of 10, 5, and 2.5 mm under tension at strain rates of $10^3$, $10^2$, and 0.1 s$^{-1}$ (bottom insert).

For smooth specimens and specimens with notch radii of 5 and 10 mm, the maximum degree of deformation was achieved in the zone of intersection of conjugated shear bands (in the center of the gauge section), which indicates ductile fracture. It is evident from Figures 2c and 3 that for these types of specimens, the ultimate strain to fracture did not vary very much at the effective strain rate of $10^3$.

Note that for the notched specimen of 2.5 mm in radius, the character of the strain distribution in the gauge section changed: the maximum strain was realized in the zone of the stress concentrator. This indicated a transition from ductile fracture to quasi-brittle fracture.

Figure 4 shows photographs of the crack formation zones in flat Ti-6Al-4V specimens and specimens with a notch ($\eta = 0.4973$) after tension at a strain rate of $10^2$ s$^{-1}$.

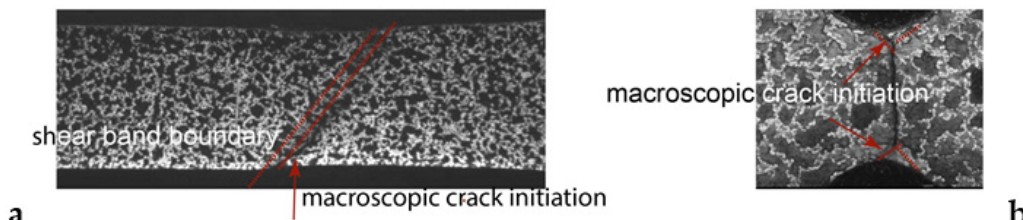

**Figure 4.** Crack in a smooth specimen after tension at a strain rate of $10^2$ s$^{-1}$ (**a**), crack in a notched specimen (**b**).

The orientation of the conjugated shear bands in smooth and notched specimens was different. The orientation of the stationary macroscopic localization bands coincided with the orientation of the forming cracks. The results obtained indicated that damage nucleated within the shear band at a scale level comparable to the grain size. Damage coalescence upon further loading led to the formation of cracks.

The intensification of the localization with increasing strain rate resulted in an increasing equivalent strain observed in the center of the specimen. Depending on the strain rate, there was a significant discrepancy between the values of the true strains and relative elongations averaged over the gauge section.

Thus, the rate of preliminary deformation and the type of stress state can be attributed to important factors in the prehistory of loading, which determine the development of fracture under tension.

### 3.2. Analysis of the Topology of the Fracture Surface

The Keyence VHX-600D digital microscope (Keyence Corporation, Osaka, Japan) was used to observe the surface topography and quantify the roughness of the fracture surface.

$S_a$ (arithmetic mean height) is the mean difference in height from the mean plane. The roughness parameter $S_a$ was determined from the 3D images of the fracture surface by the relation:

$$S_a = \frac{1}{MN} \sum_{i=1}^{N} \sum_{j=1}^{M} |h(x_i,\ y_j)|, \tag{9}$$

where $h(x_i,\ y_j)$ is the height of the relief, $x_i,\ y_j$ are the coordinates of discrete points in the projection onto the OXY plane of three-dimensional images of the crack surface, $1 \le i \le M$, and $1 \le j \le N$.

Parameter $S_z$ indicates the maximum height values $h(x_i,\ y_j)$.

The relief orientation is characterized by the numerical value of the *Str* parameter, which is estimated by the angular chart. The values of *Str* < 0.3 correspond to an anisotropic surface. *Str* > 0.5 represents an isotropic surface.

The *Spc* parameter represents the mean curvature radius of peaks for surface features categorized into peaks within the surface image. As the *Spc* value increases, the curvature of peaks grows smaller (sharper), and the curvature increases (obtuse) as the value decreases.

*Sdr* is a parameter indicating the gradient of peak growth in the superficial area.

$$Sdr = \frac{1}{A_{fs}} \iint\limits_{A_{fs}} \left[ \sqrt{1 + \left(\frac{\partial h(x,\ y)}{\partial x}\right)^2 + \left(\frac{\partial h(x,\ y)}{\partial y}\right)^2} \right] dxdy, \tag{10}$$

where $A_{fs}$ is the area of projection of the fracture surface onto the OXY plane.

The topological parameters of the fracture surfaces are shown in Table 1.

**Table 1.** Topological parameters of fracture surfaces of smooth and notched Ti-6Al-4V specimens after high-strain-rate tension.

| Dimensions $A_{fs}$, µm × µm | Parameter Values | $S_a$, µm | $S_z$, µm | *Str* | *Spc*, 1/mm | *Sdr* | Strain Rate, s⁻¹ | Notch Radius of Specimen, mm |
|---|---|---|---|---|---|---|---|---|
| 754.326 × 5422.843 | Max | 93.956 | 548.295 | 0.671 | 10,559.975 | 6.601 | 1000 | Smooth |
| | Min | 38.599 | 287.055 | 0.514 | 9296.000 | 4.771 | | |
| | Average | 66.277 | 417.675 | 0.593 | 9927.987 | 5.686 | | |
| | Std. Dev | 27.678 | 130.620 | 0.078 | 631.987 | 0.9147 | | |
| 647.931 × 3760.711 | Max | 92.319 | 522.629 | 0.128 | 14,336.275 | 12.67 | 1000 | 10 |
| | Min | 33.180 | 188.105 | 0.113 | 12,583.356 | 8.555 | | |
| | Average | 60.055 | 370.053 | 0.118 | 13,613.336 | 10.90 | | |
| | Std. Dev | 24.443 | 138.139 | 0.007 | 747.836 | 1.730 | | |
| 614.847 × 3812.108 | Max | 97.991 | 575.645 | 0.387 | 24,276.398 | 29.61 | 1000 | 5 |
| | Min | 53.723 | 405.401 | 0.080 | 18,001.793 | 18.87 | | |
| | Average | 75.598 | 511.757 | 0.188 | 21,795.751 | 25.38 | | |
| | Std. Dev | 18.076 | 75.712 | 0.140 | 2724.733 | 4.669 | | |
| 308.593 × 3842.977 | Max | 57.030 | 390.964 | 0.062 | 21,810.949 | 25.86 | 1000 | 2.5 |
| | Min | 31.290 | 215.225 | 0.048 | 15,847.240 | 13.77 | | |
| | Average | 42.736 | 316.093 | 0.057 | 19,288.885 | 20.88 | | |
| | Std. Dev | 10.700 | 74.063 | 0.006 | 2520.010 | 5.162 | | |

**Table 1.** *Cont.*

| Dimensions $A_{fs}$, μm × μm | Parameter Values | $S_a$, μm | $S_z$, μm | *Str* | *Spc*, 1/mm | *Sdr* | Strain Rate, s⁻¹ | Notch Radius of Specimen, mm |
|---|---|---|---|---|---|---|---|---|
| 292.560 × 3819.958 | Max | 261.051 | 979.485 | 0.410 | 21,296.474 | 23.75 | 100 | Smooth |
| | Min | 201.801 | 817.840 | 0.316 | 17,829.274 | 17.67 | | |
| | Average | 236.997 | 907.937 | 0.361 | 19,751.918 | 21.11 | | |
| | Std. Dev | 25.440 | 67.282 | 0.039 | 1440.505 | 2.545 | | |
| 1367.196 × 3800.668 | Max | 127.772 | 774.560 | 0.470 | 20,955.892 | 24.26 | 100 | 10 |
| | Min | 111.355 | 648.745 | 0.453 | 17,592.438 | 17.80 | | |
| | Average | 117.977 | 692.050 | 0.464 | 19,819.734 | 21.96 | | |
| | Std. Dev | 7.068 | 58.367 | 0.008 | 1575.044 | 2.947 | | |
| 174.703 × 2609.916 | Max | 255.927 | 964.900 | 0.506 | 22,991.209 | 26.68 | 100 | 5 |
| | Min | 212.121 | 831.245 | 0.424 | 20,005.690 | 22.05 | | |
| | Average | 233.770 | 885.752 | 0.472 | 21,783.518 | 24.79 | | |
| | Std. Dev | 17.887 | 57.279 | 0.035 | 1283.777 | 1.988 | | |
| 623.559 × 4957.359 | Max | 225.794 | 875.120 | 0.447 | 17,426.704 | 17.47 | 100 | 2.5 |
| | Min | 104.816 | 631.075 | 0.346 | 15,618.083 | 14.62 | | |
| | Average | 163.100 | 735.469 | 0.404 | 16,427.935 | 16.11 | | |
| | Std. Dev | 45.400 | 96.833 | 0.041 | 735.078 | 1.030 | | |

Figure 5 shows fracture surface roughness profiles of Ti-6Al-4V notched and smooth specimens after tensile tests at strain rates of $10^2$ and $10^3$ s⁻¹. Surface profiles in Figure 5 indicate that an increase in the stress triaxiality parameter from 0.33 to 0.5 led to an increase in the roughness parameter *Sa*.

The relief of the fracture surface indicates a change in the trajectory of the main crack at the final stage of fracture. This confirms that at the initial stage of fracture, microdamages originated near the bands of plastic strain localization. The orientation angles of these shear bands to the direction of macroscopic tension affected the trajectory of crack propagation.

The value of the parameter *Str*, which characterizes the asymmetry of the peaks in the relief of the fracture surface, decreased to values below 0.3 as the strain rate increased from $10^2$ to $10^3$ s⁻¹. This confirms the anisotropic nature of fracture at rates close to $10^3$ s⁻¹. The data presented in Table 1 show that the *Str* decreased with $\eta$, which indicates an increase in the contribution of directional modes in the formation of the fracture zone. The change in the topological parameters implies the possibility of embrittlement during high-speed tension at high stress triaxiality. Note that the embrittlement effect was less significant under tension at strain rates of $10^2$ s⁻¹.

The fractal dimension of the fracture surface enables us to evaluate the contribution of mechanisms of ductile and brittle fracture [23–29]. Liang et al. showed a linear relationship between the strain to fracture and the fractal dimension of the fracture surface for magnesium alloys, which belong to the same isomechanical group as Ti-6Al-4V [28].

Fractal dimensions were used to describe the degree of tortuosity on boundaries of the fracture surface:

$$D_f = 1 - \frac{log(\lambda/\lambda_0)}{logs}, \tag{11}$$

where $D_f$ is the fractal dimension of the fracture surface, $\lambda$ is the measured length, $\lambda_0$ is a constant (Euclidean length), and $s$ is the measurement scale [30].

The fractal dimension $D_f$ increased linearly with $S_a$, which could be described by the following Equation (12) [26]:

$$D_f = D_0 + k_1 S_a, \tag{12}$$

where $S_a$ is the surface roughness, and $D_0$ and $k_1$ are coefficients.

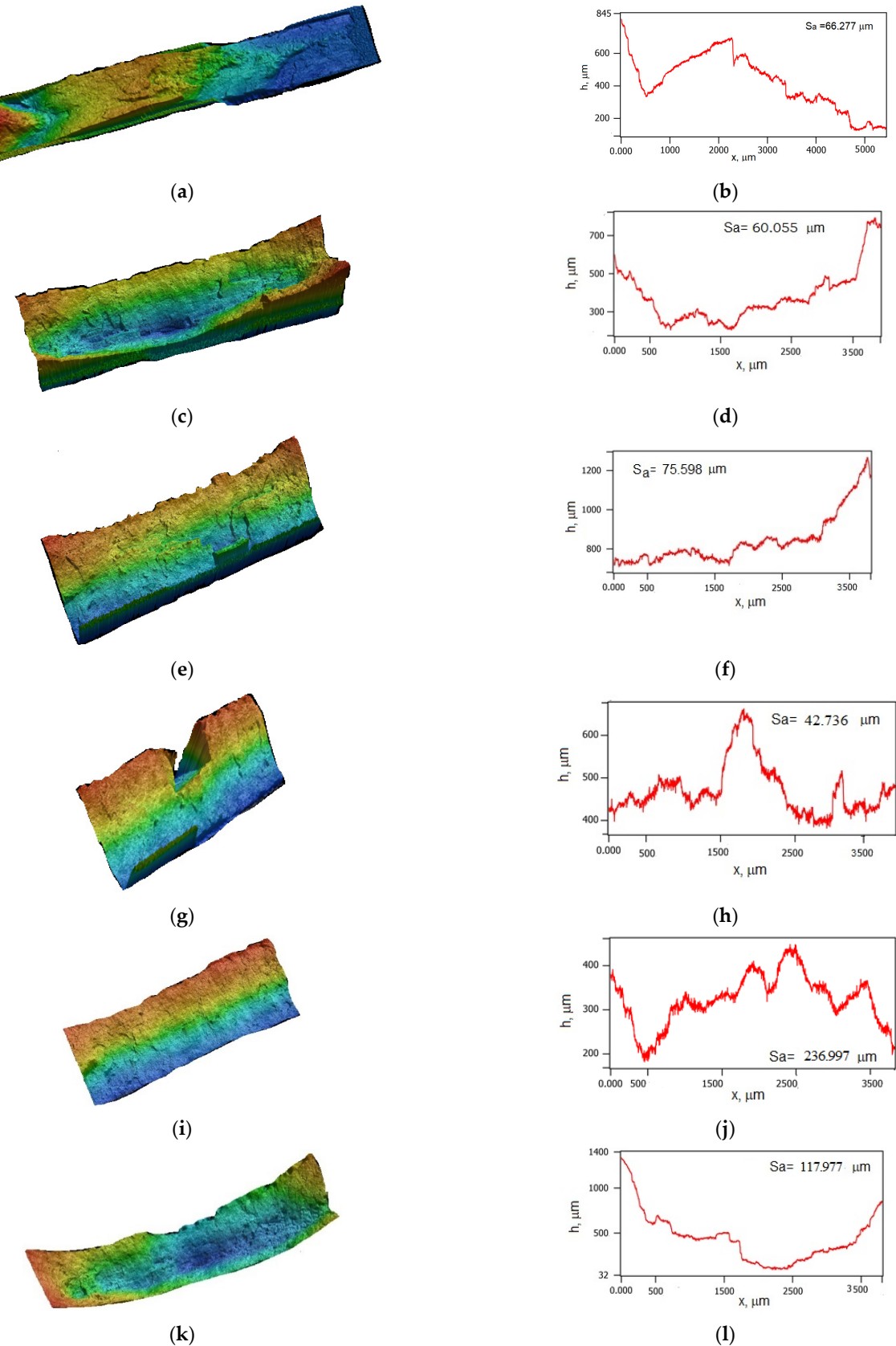

**Figure 5.** *Cont.*

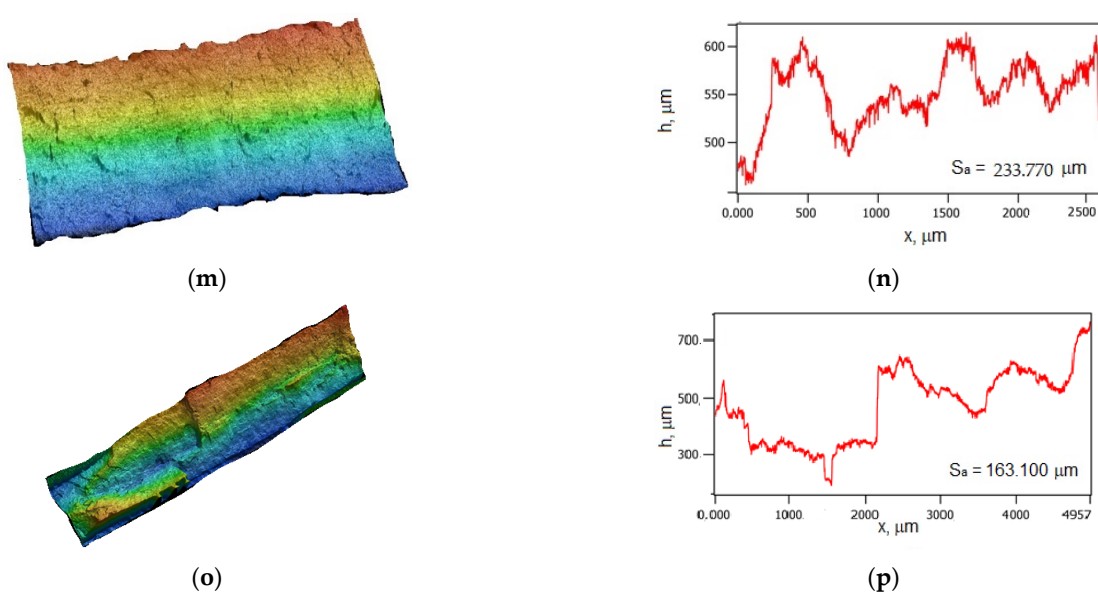

**(m)**

**(n)**

**(o)**

**(p)**

**Figure 5.** Roughness profiles of Ti-6Al-4V fracture surfaces. Left column shows 3D images of the fracture surface profile; right column shows profiles of height *h* along a straight-line segment parallel to the OX axis and equidistant from the surfaces of the specimen along a line equidistant from the sample surfaces. Strain rate and notch radius are: $10^3$ s$^{-1}$, smooth (**a,b**); $10^3$ s$^{-1}$, 10 mm (**c,d**); $10^3$ s$^{-1}$, 5 mm (**e,f**); $10^3$ s$^{-1}$, 2.5 mm (**g,h**); 100 s$^{-1}$, smooth (**i,j**); 100 s$^{-1}$, 10 mm (**k,l**), 5 mm (**m,n**); 100 s$^{-1}$, 2.5 mm (**o,p**).

The values of coefficients $D_0$ = 2.59 and $k_1$ = −0.0001353 1/μm were estimated from the averaged data on the fracture surface roughness at a strain rate of $10^2$ s$^{-1}$. The values of $D_0$ = 2.51 and $k_1$ = −0.0001585 1/μm were estimated at a strain rate of $10^3$ s$^{-1}$. A decrease in the fractal dimension indicates a tendency toward embrittlement of Ti-6Al-4V with increasing strain rates. It is evident from Figure 5 that the contribution of a cleavage mode of fracture increased with the strain rate. However, intensive heating within the localized shear band, which took place at $10^3$ s$^{-1}$, prevented the nucleation of voids required for cleavage to occur, and led to an overall higher fracture energy at high strain rates than those at lower strain rates, as shown in Figure 6. The maximum and average values of *Sa* changed with $\eta$. This indicates that stress triaxiality affected not only void growth rate but also void nucleation.

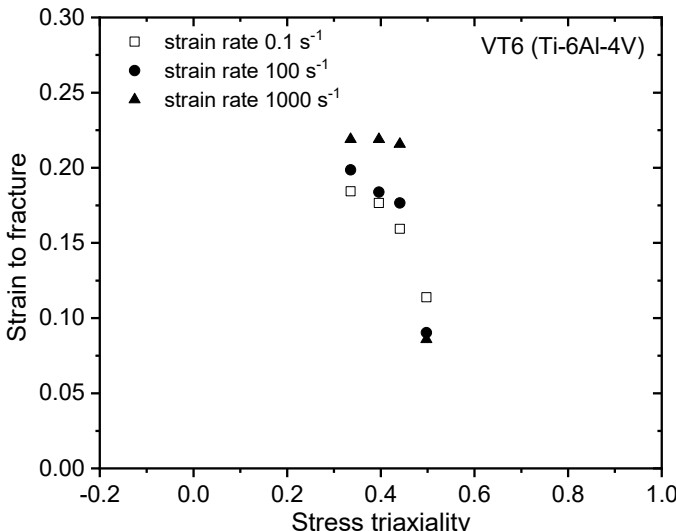

**Figure 6.** Strain to fracture of Ti-6Al-4V versus stress triaxiality at tension strain rates of 0.1, $10^2$, and $10^3$ s$^{-1}$.

## 4. Discussion

DIC analysis was performed in this work to investigate the evolution of plastic strain and fracture formation in Ti-6Al-4V under a range of strain rates from 0.1 to $10^3$ s$^{-1}$ and a range of stress triaxialities from 0.33 to ~0.5. At high strain rates, it was found that the fracture of Ti-6Al-4V was initiated on boundary edges of the localized bands, while the highest magnitude of plastic strain was realized in the zone of shear band intersection. This implies that the initiation of fracture occurred in the zones of high gradients of deformation and shear stresses.

It is evident from Figure 4 that the orientation of the stationary macroscopic localization bands coincided with the orientation of the forming cracks. The results obtained confirmed the difference between structural changes due to the formation of shear bands in the Ti-6Al-4V alloy and alpha titanium alloys. The difference between the structural changes in alpha and alpha + beta titanium alloys was pointed out in [5].

The analysis of the fracture surface topology showed that the contribution of the quasi-cleavage mode increased with strain rate. It is evident from Figure 5 that fracture surfaces after tension at $10^3$ s$^{-1}$ were more irregular and tortuous than those after tension at $10^2$ s$^{-1}$. The crack deflection took place during fracture of the notched specimen with a radius of 2.5 mm at $10^2$ s$^{-1}$ and all types of specimen at $10^3$ s$^{-1}$. The observed decrease in fractal dimension $D_f$ (Equation (12)) with increasing strain rate confirmed the changes in the mechanisms of fracture at the mesoscopic level. These findings correlate with the results of [12].

Note that the indicated tendency to embrittlement with increasing strain rate was not observed in an alpha titanium alloy VT5-1 (analog Ti-5Al-2.5Sn) [18].

Thus, the laws of damage initiation and crack growth in alpha and alpha + beta titanium alloys are different under high strain rate. These differences must be taken into account when predicting the dynamic fracture of critical titanium structures. This conclusion is consistent with the results of SEM studies of the fracture surface of Ti-6Al-4V presented in [31]. It was shown that damage, in the form of micro-cavities, nucleated mostly at $\alpha/\beta$ interfaces. The microstructure of alpha-titanium alloys, for example, Ti-5Al-2.5Sn, lacks $\alpha/\beta$ interfaces. Thus, the results obtained in this work should not be generalized for the cases of deformation and fracture of one-component HCP alloys.

In this work, we used specimens cut along the rolling direction. It was shown in [32] that the rolled Ti-6Al-4V may exhibit significant plastic deformation anisotropy, which must be taken into account when calibrating constitutive equations and fracture criteria. However, the equiaxed grain structure is indirect evidence that the alloy under investigation did not have significant plastic deformation anisotropy. Thus, data presented in Figure 6 can be used to calibrate damage and fracture models.

## 5. Conclusions

The experimental results obtained in this paper can be summarized as follows:

(1)  The formation of shear bands at high strain rates influences the fracture behavior of the Ti-6Al-4V alloy.
(2)  The analysis of the fracture surface indicates the activation of a quasi-cleavage mode of fracture at high strain rates in the Ti-6Al-4V alloy.
(3)  At relatively low values of stress triaxiality η and at high strain rates, the ultimate strain to fracture appears to change insignificantly.
(4)  High stress triaxiality can cause embrittlement of Ti-6Al-4V subjected to high strain rates.
(5)  At high strain rates, the Ti-6Al-4V alloy undergoes fracture by cleavage formation followed by coalescence of nucleated microvoids.

The data obtained expand the understanding of the regularities of the influence of the stress triaxiality on shear instabilities occurring at strain rates up to $10^3$ s$^{-1}$. Such strain rates can be realized in a number of technological processes for the production of structural elements, which are designed by means of CAM/CAE (Computer Aid Manufacturing/Computer Aid Engineering). Accounting for the stress triaxiality is essential to

improve cutting and drilling technologies of titanium alloys because of the complex stress state in the chip formation zone.

**Author Contributions:** Conceptualization, V.V.S. and V.A.S.; methodology, V.V.S. and V.A.S.; formal analysis, V.V.S.; investigation, V.V.S. and V.A.S.; resources, V.A.S.; data curation, V.V.S.; writing—original draft preparation, V.V.S. and V.A.S.; writing—review and editing, V.V.S. and V.A.S.; visualization, V.V.S.; supervision, V.A.S.; project administration, V.V.S.; funding acquisition, V.V.S. All authors have read and agreed to the published version of the manuscript.

**Funding:** This research was funded by the Russian Science Foundation (RSF), grant No. 20-79-00102.

**Data Availability Statement:** Data obtained within this research and supporting reported results are referenced in the report on grant No. 20-79–00102.

**Acknowledgments:** The authors are grateful for the support of this research. Authors thank A.A. Kozulin, J. Starcevich, and A.V. Chupashev for the help in experimental tests.

**Conflicts of Interest:** The authors declare no conflict of interest.

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
