# Peer review of "Localization of Plastic Deformation in Ti-6Al-4V Alloy"

_metals, doi:10.3390/met11111745_

Round 1
Reviewer 1 Report
The abstract is too verbose. Please use a clear, concise, convincing language to write the abstract, including context, methods, results and conclusion。
Please see the attachment.

Author Response
Authors thank the reviewer for valuable comments. Please, find the word document with the response to the comments in the attachment

Reviewer 2 Report
This manuscript (metals-1424666-peer-review-v1) experimentally and theoretically investigated the influencing factors of stress, strain, microstructure and fracture surface etc. on plastic flow and deformation of Ti-6Al-4V titanium alloy under tension.
The research background was well elaborated, and the research purpose was adequately proposed. The experiment was systematic and complete, whose methodology was clearly and professionally described. The figures and tables are eloquent and supportive for the explanation of the results, and the conclusions are concise and in close correlation with the results. Besides, the English expression throughout the manuscript was good enough.
To sum up, the research has a high level of originality and quality, I have no further suggestions for making any modification of it, and therefore suggest it be accepted by the journal Metals in its current form.
Author Response
Authors thank the reviewer for appreciating the manuscript

Reviewer 3 Report
- The used material is VT6, not Ti-6Al-4V, although they have similar composition. Please revised the name of the material in the title and text of this manuscript.
- English is poor, some sentence is hard to understand. So, the manuscript should extensively polished.
- Page 2, “In [6], results indicate like melting zone in localized shear bands in the Ti-6Al-4V alloy under dynamic compression at an 41 initial temperature of 295 K.”. What does it mean?
- Literature review about the deformation, strain localization, and facture behavior of titanium alloy under high strain rate is not enough. Please add some related reference about these topics.
- The authors said “The aim of this work was to obtain new experimental data on the laws governing the development of plastic flow 53 and deformation of the Ti-6Al-4V alloy under tension in the range of strain rates from 0.1 to 103 s-1.”. The research aim needs refining. The aim should not to be the acquisition of the new experiment data but to reveal some unclear mechanism, etc.
- Fig 1a, please give some marks in the figure to indicate which is alpha phase, which is beta phase.
- Page 4, equation number is error.
- Page 6, DIC image is too small.
- Page 6, “The results obtained indicate that in the zone of formation of bands of macroscopic localization of plastic deformation, damages arise at a lower scale level comparable to the 174 grain size.”. What does it mean, and how to explain the relationship between the strain band and the damage?. It seems that there are no explicit evidence in Fig 3 that support this argument.
- Section 3.2, the author should firstly provide the corresponding parameters charactering the fracture surface, and then analyze the experimental results.
Author Response
Authors thank the reviewer for valuable comments. Please, find the response to the comments in the attachment.

Reviewer 4 Report
- This paper may be interesting to specialists due to a wide range of tensile stress rates used for deformation tests of VT6 Titanium alloy. The results on localization of plastic flow and on damages/fracture formation at high stress rates are new and useful to better understanding of a "quasy - brittle" behavior of Titanium alloy at fast deformation conditions. The critical remark concerns the lack of information on grain structure evolution from initial state (Fig. 1,a) to ultimate elongation before the fracture of samples. At high stress rates there is a competition between dynamic of recrystallization and damages formation observed in the experiment. These kind of data would be very useful to understand the mechanism of plastic deformation.
- The Part 3.2 of the paper which represent the fractographic analysis of fracture surfaces, looks a "stranger" in this paper. The Fig. 5 together with Table 1 are not representative enough to improve the understanding deformation and fracture experimental results. At least, this Part may be shorten if exclude Fig. 5 and Table 1.
Author Response

(The authors gave the same response as above.)

Round 2
Reviewer 1 Report
The authors have made very good progress on improving this manuscript.Reviewer 3 Report
none